

# Physical and mental effects of different radical prostatectomy techniques on urologic surgeons

Mahmut T. Olcucu[1], Mustafa S. Bolat[2], Kadir Yildirim[3], Yasar Ozgok[4], Theodoros Tokas[5,6] and Ali Gözen[7,†]

[1] Department of Urology, University of Health Sciences, Antalya Training and Research Hospital, Antalya, Turkey
[2] Department of Urology, Atilim University, Ankara, Turkey
[3] Department of Urology, Malatya Turgut Ozal University, Malatya, Turkey
[4] Department of Urology, Ankara Yuksek Ihtısas University, Ankara, Turkey
[5] Department of Urology and Andrology, General Hospital Hall i.T, Hall in Tirol, Austria
[6] Department of Urology, A part of European Urology, Training and Research in Urological Surgery and Technology (T.R.U.S.T.)-Group, Amsterdam, Netherlands
[7] Department of Urology, Medius Kliniken, University of Tubingen, Stuttgart, Germany
[†] Deceased.

Corresponding author
Mahmut T. Olcucu,
matah_ol@hotmail.com

## ABSTRACT

**Objective**. In this web-based international survey study, we aimed to show an association between physical exhaustion and patient, relatives, and employer-related mental stress for surgeons performing open radical prostatectomy (ORP), laparoscopic radical prostatectomy (LRP), and robot-assisted laparoscopic prostatectomy (RALP). Additionally, we also aimed to compare the outcomes of three approaches.

**Methods**. We sent a ten-question survey to the urologists performing ORP, LRP, and RALP *via* e-mail and social media. Only fully completed surveys were included in the study analysis. We asked questions about age, the preferred surgical approach for radical prostatectomy, frequency of weekly exercise, and their possible associations with physical exhaustion and musculoskeletal complaints.

**Results**. A total of 160 urologists completed the survey. The RALP group showed a lower physical exhaustion rate and increased eye strain ($p < 0.001$) and $p = 0.002$, respectively). Although walking was the most preferred sports activity, no correlation was found between regular sport or exercise and musculoskeletal complaints ($p > 0.05$).

**Conclusion**. Compared to ORP and LRP, physical exhaustion was lower in the RALP technique. Although the number of participants was limited, regular exercise weakly improved physical exhaustion and musculoskeletal complaints. We believe that regular sports activities by urologists dealing with LRP and RALP will help relieve physical discomfort.

# INTRODUCTION

Radical prostatectomy is the primary surgical treatment for prostate cancer. Open anatomic radical prostatectomy (ORP) and laparoscopic radical prostatectomy (LRP) were

introduced by *Walsh & Donker (1982)*; *Schuessler et al. (1997)*, respectively. Compared to open surgery, laparoscopic procedures offer several advantages, including reduced postoperative pain, shorter hospital stays, faster recovery, and improved cosmetic outcomes. However, laparoscopic surgery can impose significant physical and mental stress on surgeons. The prolonged immobility of the wrist, head, and neck—often due to the surgeon's position relative to imaging systems—can result in considerable strain. The introduction of robot-assisted laparoscopic prostatectomy (RALP) in the early 21st century addressed many of these challenges (*Gofrit et al., 2008*; *Hemal, Srinivas & Charles, 2001*; *Pasticier et al., 2001*). RALP provides ergonomic benefits, such as improved manipulation *via* three-dimensional imaging and enhanced operating posture, allowing the surgeon to work from a seated position, unlike LRP and ORP (*Bagrodia & Raman, 2009*; *Andolfi et al., 2019*).

Despite technological advancements, certain factors can still lead to suboptimal conditions during radical prostatectomy. Stress can stem from patient expectations, family pressures, and institutional demands, often creating psychological burdens for surgeons (*Bolat et al., 2019*). Regular physical exercise has well-documented physical and mental health benefits. Prior research has examined the impact of exercise on surgeons' musculoskeletal strength and ergonomics in the operating room (*Winters et al., 2020*). In some cultures, the expectation of flawless outcomes in minimally invasive surgeries, coupled with an intolerance for complications, contributes further to the stress experienced by surgeons (*Waljee et al., 2014*). Moreover, numerous studies have shown that regular exercise helps prevent chronic illnesses such as cardiovascular diseases and cancers, while also alleviating musculoskeletal pain (*Wewege et al., 2018*; *Cheville, Smith & Basford, 2018*; *Rosenbaum & Sherrington, 2011*).

In this web-based international survey study, we aimed to investigate the association between physical exhaustion and mental stress related to patients, their relatives, and employers among surgeons performing ORP, LRP, and RALP.

## MATERIALS AND METHODS

Some content in this manuscript was previously published in a preprint version (DOI: 10.21203/rs.3.rs-3976248/v1).

After obtaining approval from the local ethics committee of Antalya Training and Research Hospital (Approval No: 2019-023), we developed a survey using the SurveyMonkey platform (http://www.SurveyMonkey.com). The survey, along with informed consent forms, was distributed *via* email and social media platforms to urologists who perform radical prostatectomies. Participants were recruited through databases provided by the Turkish Association of Urology's members.

Complaint severity was assessed using a numerical rating scale ranging from 0 (no symptoms) to 10 (worst imaginable symptoms). Only fully completed surveys were included in the statistical analysis. The survey included the following questions regarding various radical prostatectomy techniques (see Files S1):

Q1: Preferred technique among robotic, laparoscopic, and open radical prostatectomy, and frequency of use over a specific period

Q2: Patient-related stress

Q3: Employer-related stress

Q4: Sources of stress experienced during surgery

Q5: Physical exhaustion associated with each prostatectomy technique

Q6: Musculoskeletal complaints during or after surgery

Q7: Current physical complaints influencing the choice of prostatectomy technique

Q8: Need for medical support for musculoskeletal issues

Q9: Information regarding regular exercise and sports activities

Q10: Demographic data

This study also aimed to assess whether physical activity has a beneficial effect on musculoskeletal complaints among surgeons performing radical prostatectomies. We hypothesized that robot-assisted laparoscopic prostatectomy (RALP) would be associated with less physical exhaustion and fewer musculoskeletal complaints compared to other techniques. Our second hypothesis was that urologists who engage in regular physical exercise would report fewer musculoskeletal complaints.

### Statistical analysis

We used SPSS Statistics for Windows, Version 25.0 (IBM Corp., Armonk, NY, USA) for all analyses. Data were expressed as $n$ (%) for categorical variables, and as mean $\pm$ standard deviation (SD) or median (minimum–maximum) for continuous variables, as appropriate. The Shapiro–Wilk test was used to assess the normality of data distribution. For comparisons, we used one-way ANOVA for parametric variables and the Kruskal–Wallis test for non-parametric variables. The Bonferroni-Dunn test was employed as a *post-hoc* analysis for significant findings. Spearman's rank correlation test, adjusted for age and body mass index (BMI), was used to evaluate correlations between variables. A *p*-value <0.05 was considered statistically significant.

## RESULTS

A total of 160 certified urologists working in private practice, state, or university hospitals completed the survey. Among the participants, 127 (79.37%) reported performing ORP, 58 (36.25%) performed LRP, and 37 (23.12%) performed RALP. Of these, 75 (46.87%) exclusively performed ORP, 17 (10.62%) performed LRP, and 10 (6.25%) exclusively performed RALP. The majority of participants had surgical experience ranging from 1 to 100 procedures for all three techniques: 98.42% for ORP, 93.10% for LRP, and 91.89% for RALP ($p = 0.09$). A total of 150 participants (93.70%) reported experiencing stress due to patient attitudes and behaviour, while 96 participants (60.0%) cited stress originating from employers The most common patient-related stressors identified were morbid obesity (71.25%) and a history of previous abdominal surgery (51.87%). Ninety-eight participants (58.0%) reported no prior professional support for their complaints, and 26 participants (16.25%) stated they had never changed their surgical technique (Table 1). There were no significant differences in mean age or BMI across the groups based on the preferred surgical technique ($p > 0.05$). Surgeons performing RALP reported significantly lower levels of physical exhaustion compared to those performing LRP and ORP ($p < 0.001$).

However, the RALP group reported a higher incidence of eye strain ($p = 0.002$). All other complaints were comparable across groups ($p > 0.05$) (Table 2). One hundred and fifteen surgeons (71.87%) engaged in some form of physical activity, with walking being the most commonly preferred (Table 3 and Table S1).

## DISCUSSION

Many factors can contribute to stress among surgeons, including patient-related, work-related, social, and personal issues (*Arora et al., 2010*). In the modern era, urologists who are exposed to minimally invasive techniques early in their residency often face a shorter learning curve for robotic and laparoscopic procedures. In contrast, urologists trained primarily in open surgery may experience a longer and more challenging adaptation period to meet the evolving demands of contemporary surgical practice. Minimally invasive procedures, while beneficial for patients, can be a significant source of stress for some surgeons. It is well-established that heightened stress can negatively impact surgical outcomes. In contrast, effective stress management can enhance surgical performance (*Cheville, Smith & Basford, 2018*). A study from Turkey reported that a gradual increase in occupational stress can lead to psychological deterioration and burnout syndrome, a condition commonly observed among Turkish urologists (*Bolat et al., 2019*). While minimally invasive techniques offer clinical benefits to patients, they do not always ensure ergonomic ease for surgeons. For instance, conventional laparoscopy lacks tactile feedback and provides only a two-dimensional view, potentially prolonging surgical time and contributing to ergonomic discomfort. In contrast, robotic-assisted surgery enables more complex procedures through enhanced visualization, improved access, and three-dimensional maneuverability—offering superior ergonomic conditions, especially in experienced hands (*Renda & Vallancien, 2003*). Morbid obesity presents a particular challenge, increasing the likelihood of intraoperative and postoperative complications and thus elevating stress levels among surgeons. Consistent with previous studies, our findings indicate that 71.25% of participating surgeons experience increased stress when operating on obese patients undergoing radical prostatectomy. *Eden et al. (2006)* reported that although obesity may increase operative time in laparoscopic prostatectomy, it does not significantly affect other intraoperative or postoperative parameters. This discrepancy may be attributed to the extensive surgical experience of the surgeons in their study.

Our findings regarding the relationship between physical complaints and exercise suggest that regular physical activity may alleviate certain musculoskeletal symptoms, including elbow stiffness, hand and leg pain, and finger numbness—although the number of participants reporting these improvements was limited. Laparoscopy is often associated with suboptimal static posture. A study evaluating the effects of the ETHOS surgical chair, three-dimensional imaging technology, and Radius Surgical System manipulators on surgical time and surgeon comfort found that these innovations provided comfort and operative efficiency comparable to the da Vinci robotic system, outperforming traditional laparoscopic methods (*Tokas et al., 2017*). While some reports describe RALP as a user-friendly platform due to its three-dimensional visual interface, *Lee et al. (2017)* found

**Table 1 Rate of responses to questions 1–4 and questions 7 and 8.**

**Q1. Annual number of radical prostatectomy (n = 222)**

| | Response | n (%) |
|---|---|---|
| **Open radical prostatectomy (n = 127)** | 1–100 | 125 (56.3) |
| | 100–200 | – |
| | 200–300 | 2 (0.9) |
| | >300 | – |
| | **Response** | |
| **Laparoscopic radical prostatectomy (n = 58)** | 1–100 | 54 (24.3) |
| | 100–200 | 3 (1.4) |
| | 200–300 | 1 (0.5) |
| | >300 | – |
| | **Response** | |
| **Robot-assisted radical prostatectomy (n = 37)** | 1–100 | 34 (15.3) |
| | 100–200 | 2 (0.9) |
| | 200–300 | – |
| | >300 | 1 (0.5) |
| | **Response** | ***n*, (%)** |
| **Q2. Feeling of stress arising from patients and/or relatives prior radical prostatectomy (n = 160)** | Always | 13 (8.1) |
| | Usually | 49 (30.6) |
| | Sometimes | 56 (35) |
| | Rarely | 32 (20) |
| | Never | 10 (6.3) |
| | **Response** | ***n*, (%)** |
| **Q3. Feeling of stress due to increased expectations of employer (n = 160)** | Always | 3 (1.8) |
| | Usually | 14 (8.8) |
| | Sometimes | 32 (20) |
| | Rarely | 47 (29.4) |
| | Never | 64 (40) |
| | **Response** | ***n*, (%)** |
| **Q4. What causes stress during radical prostatectomy? (multiple choice avaliable) (n = 426)** | None | 1 (0.2) |
| | Patient with morbid obesity | 114 (26.7) |
| | Grade or stage of the disease | 68 (16) |
| | Prior history of abdominal operation | 83 (19.5) |
| | Frequent change of the surgical team | 31 (7.3) |
| | Complications during or after the surgery | 87 (20.4) |
| | Lack of standard postoperative care | 17 (4) |
| | Postoperative follow-up | 17 (4) |
| | Other | 8 (1.9) |
**Table 1** (*continued*)

| | Response | n (%) |
|---|---|---|
| | No | 98 (58) |
| | Lifestyle modifications | 29 (18) |
| **Q7. Getting professional support regarding complaints (multiple choice) (n = 169)** | Physical therapy modalities (Massage, TENS, dry needling, hot-cold compress, stretching) | 22 (13) |
| | Medical treatment(s) | 13 (7.7) |
| | Surgery | 4 (2.4) |
| | Other | 3 (1.8) |
| | **Response (n = 133)** | **n (%)** |
| | Always | 1 (0.8) |
| **Q8. Do your current complaints affect choice of radical prostatectomy technique? (n = 133)** | Usually | 14 (10.4) |
| | Sometimes | 9 (6.8) |
| | Rarely | 58 (43.6) |
| | Never | 26 (19.5) |
| | I can perform only one type of technique | 25 (18.8) |

**Notes.**

BMI, body-mass index; SD, standard deviation; TENS, transcutaneous electrical nerve stimulation.

that 56.1% of surgeons still experienced physical discomfort during robotic procedures. Similarly, *Tokas (2020)* noted that laparoscopic radical prostatectomy (LRP) caused discomfort primarily in the upper back, shoulders, arms, and wrists, while RALP was more often associated with strain in the forehead, neck, and trunk. These reports support our hypothesis that RALP may lead to less physical exhaustion and fewer musculoskeletal complaints compared to other techniques (*Rassweiler et al., 2010*). Among the three main approaches—open, laparoscopic, and robot-assisted radical prostatectomy—RALP appears to be associated with the least physical exhaustion. This may be due to the procedure being performed in a seated position, the provision of a three-dimensional surgical view, and the enhanced comfort and precision of hand and wrist movements (*Andolfi et al., 2019*; *Dalager et al., 2017*). A comparative study of robotic and conventional laparoscopic surgery further confirmed that robotic-assisted procedures, in experienced hands, offer superior ergonomics for the muscle groups most frequently engaged during surgery (*Dalager et al., 2017*). Finally, a meta-analysis examining the effects of physical exercise on the musculoskeletal system concluded that regular physical activity is beneficial for individuals with chronic musculoskeletal pain (*O'Connor et al., 2015*).

Studies investigating the learning curve for robotic and laparoscopic prostatectomy have shown that completing 30 to 50 consecutive cases is generally sufficient to gain proficiency (*Menon et al., 2002*; *Fabrizio, Tuerk & Schellhammer, 2003*). In our study, most participants reported performing approximately 100 cases annually across all three prostatectomy techniques. Consistent with previous findings, surgeons in the RALP group reported a higher incidence of eye strain, while other physical complaints remained relatively stable (*Plerhoples, Hernandez-Boussard & Wren, 2012*). Our data regarding physical strain support the ergonomic superiority of the RALP technique over laparoscopic and open approaches. *Hemal, Srinivas & Charles (2001)* found that laparoscopic surgeons frequently experienced significant ergonomic issues, particularly finger numbness and eye strain, with younger

Table 2 Demographic data and analysis of responses to questions 5 and 6 between groups.

| | Open RP (n = 127) Mean ± SD Median (min–max) | Laparoscopic RP (n = 58) Mean ± SD Median (min–max) | Robot-assisted RP (n = 37) Mean ± SD Median (min–max) | P value |
|---|---|---|---|---|
| Age (year)$^{\beta}$ | 45.83 ± 0.87 45 (29–71) | 44.24 ± 1.17 43 (29–65) | 45.70 ± 1.76 43 (29–76) | 0.555 |
| BMI$^{\beta}$ | 26.97 ± 0.97 26.89 (20.04–35.56) | 26.55 ± 0.41 26.22 (19.83–36.24) | 27.19 ± 0.59 27.47 (20.04–34.63) | 0.683 |

Q5. How much does/do the preferred technique(s) exhaust you physically?$^{\beta}$

| | Open RP (n = 127) Mean ± SD Median (min–max) | Laparoscopic RP (n = 58) Mean ± SD Median (min–max) | Robot-assisted RP (n = 37) Mean ± SD Median (min–max) | P value |
|---|---|---|---|---|
| Answers | 4.85 ± 2.32[a] 5 (0–10) | 5.10 ± 2.62[a] 5 (0–9) | 2.7 ± 1.89[b] 2 (0–8) | <0.001[*] |

Q6. Which of these complaints do you experience during or after performing radical prostatectomy with the preferred technique(s)?

| | Open RP (n = 127) Mean ± SD Median (min–max) | Laparoscopic RP (n = 58) Mean ± SD Median (min–max) | Robot-assisted RP (n = 37) Mean ± SD Median (min–max) | P value |
|---|---|---|---|---|
| Forehead pain | 0.88 ± 1.66 0 (0–8) | 1.17 ± 2.18 0 (0–9) | 1.54 ± 2.30 1 (0–9) | 0.108 |
| Eye strain | 0.82 ± 1.63[a] 0 (0–8) | 1.53 ± 2.22[a, b] 0 (0–8) | 2.16 ± 2.58[b] 0 (0–9) | 0.001[*] |
| Neck pain | 2.17 ± 2.72 0 (0–10) | 2.21 ± 2.61 0 (0–9) | 2.19 ± 2.50 1 (0–8) | 0.861 |
| Back pain | 2.53 ± 2.80 2 (0–10) | 2.83 ± 2.78 3 (0–10) | 1.84 ± 2.67 0 (0–9) | 0.170 |
| Shoulder stiffness | 1.20 ± 2.33 0 (0–10) | 1.79 ± 2.62 0 (0–10) | 1.35 ± 2.12 0 (0–9) | 0.252 |
| Chest pain | 0.39 ± 1.08 0 (0–7) | 0.50 ± 1.60 0 (0–8) | 0.30 ± 0.52 0 (0–2) | 0.551 |
| Arm pain | 1.50 ± 2.46 0 (0–10) | 1.74 ± 2.80 0 (0–10) | 0.81 ± 1.61 0 (0–7) | 0.436 |
| Forearm pain | 1.06 ± 2.26 0 (0–10) | 1.07 ± 2.14 0 (0–8) | 0.81 ± 1.72 0 (0–8) | 0.880 |
| Elbow stiffness | 0.51 ± 1.51 0 (0–10) | 0.90 ± 2.06 0 (0–8) | 0.89 ± 2.17 0 (0–9) | 0.432 |
| Hand pain | 1.22 ± 2.16 0 (0–10) | 0.90 ± 1.83 0 (0–7) | 1.86 ± 2.80 0 (0–8) | 0.164 |
| Wrist stiffness | 0.98 ± 2.04 0 (0–10) | 1.09 ± 2.1 0 (0–9) | 1.38 ± 2.53 0 (0–8) | 0.501 |
| Finger numbness | 1.11 ± 2.11 0 (0–8) | 1.78 ± 2.62 0 (0–10) | 1.86 ± 2.94 0 (0–9) | 0.170 |

**Table 2** (*continued*)

|  | Open RP (*n* = 127) Mean ± SD Median (min–max) | Laparoscopic RP (*n* = 58) Mean ± SD Median (min–max) | Robot-assisted RP (*n* = 37) Mean ± SD Median (min–max) | *P* value |
|---|---|---|---|---|
| **Leg pain** | 1.83 ± 2.78 0 (0–10) | 2.43 ± 2.97 1 (0–9) | 1.32 ± 2.29 0 (0–9) | 0.270 |
| **Total complaints** | 14.97 ± 27.42 10 (0–97) | 19.03 ± 18.84 14.50 (0–83) | 16.45 ± 17.87 13 (0–60) | 0.223 |

Notes.

BMI, body-mass index; RP, radical prostatectomy; SD, standard deviation.

[β] Parametric tests were used (Non-parametric tests are used in the remanings).

*Significant *p* values.

[a,b,c] Different lowercases denote statistically significant differences.

[−] Visual analog scale (0–10) was used to score for the complaints.

**Table 3  Frequency of weekly excercise in response to question 9.**

|  | 0/week *n*, (%) | 1/week *n*, (%) | 2/week *n*, (%) | 3/week *n*, (%) | 4/week *n*, (%) | 5/week *n*, (%) | 6/week N *n*, (%) | 7/week *n*, (%) |
|---|---|---|---|---|---|---|---|---|
| Walking | 45(28.12) | 10(6.25) | 21(13.12) | 29(18.12) | 13(8.12) | 16(10) | 6(3.75) | 20(12.5) |
| Running | 115(71.87) | 14(8.75) | 8(5) | 16(10) | 4(2.5) | 1(0.62) | 0(0) | 2(1.25) |
| Bicycling | 132(82.50) | 11(6.87) | 6(3.75) | 2(1.25) | 4(2.5) | 2(1.25) | 1(0.62) | 2(1.25) |
| Swimming | 136(85) | 7(4.37) | 5(3.12) | 3(1.87) | 2(1.25) | 3(1.87) | 3(1.87) | 1(0.62) |
| Football | 148(92.5) | 8(5) | 3(1.87) | 0(0) | 0(0) | 0(0) | 0(0) | 1(0.62) |
| Basketball | 151(94.37) | 4(2.5) | 4(2.5) | 0(0) | 1(0.62) | 0(0) | 0(0) | 0(0) |
| Volleyball | 156(97.5) | 2(1.25) | 2(1.25) | 0(0) | 0(0) | 0(0) | 0(0) | 0(0) |
| Tennis | 154(96.25) | 4(2.5) | 1(0.62) | 0(0) | 0(0) | 1(0.62) | 0(0) | 0(0) |
| Golf | 157(98.12) | 1(0.62) | 1(0.62) | 1(0.62) | 0(0) | 0(0) | 0(0) | 0(0) |
| Weight Lifting | 136(85) | 3(1.87) | 8(5) | 8(5) | 2(1.25) | 2(1.25) | 0(0) | 1(0.62) |
| Boxing | 154(96.25) | 1(0.62) | 3(1.87) | 2(1.25) | 0(0) | 0(0) | 0(0) | 0(0) |
| Meditation | 151(94.37) | 2(1.25) | 3(1.87) | 1(0.62) | 2(1.25) | 0(0) | 0(0) | 0(0) |
| Yoga | 153(95.62) | 2(1.25) | 2(1.25) | 2(1.25) | 1(0.62) | 0(0) | 0(0) | 0(0) |
| Pilates | 155(96.87) | 2(1.25) | 0(0) | 1(0.62) | 0(0) | 0(0) | 0(0) | 2(1.25) |

surgeons being more affected. *Matern & Waller (1999)* reported that muscle exertion during laparoscopic surgery was six times higher compared to open surgery. Despite the physical demands of laparoscopic surgery, our results showed that 63.14% of surgeons did not alter their preferred radical prostatectomy technique. Additionally, the majority were open to adopting new surgical approaches, with only 18.79% exclusively using one technique. This willingness to embrace diverse techniques may contribute to increased physical exhaustion, particularly during the initial learning period. Walking was the most commonly preferred form of physical activity among participating urologists. A slight correlation was observed between regular physical activity and reduced musculoskeletal complaints in surgeons performing radical prostatectomy (Table 3). Engaging in regular fitness activities may help mitigate or prevent musculoskeletal strain during surgery (*Schlussel & Maykel, 2019*). Some authors have suggested that consistent physical activity may protect against exhaustion (*DeHert, 2020*). A well-documented link exists between

inadequate exercise and the development of chronic diseases (*Booth, Roberts & Laye, 2012*). Furthermore, a meta-analysis concluded that regular and sufficient physical exercise can reduce musculoskeletal complaints without incurring additional treatment costs (*Miyamoto et al., 2019*).

In a web-based survey of 701 urologists worldwide, walking and running were reported as the most common forms of exercise. Consistent with those findings, our results also demonstrated that regular physical activity offers protective benefits against low back pain (*Lloyd et al., 2019*). Urologists who engage in regular exercise or sports activities reported lower levels of physical exhaustion and musculoskeletal complaints, supporting our hypothesis. We believe that encouraging urologists—particularly those performing LRP and RALP—to participate in regular physical activity may help alleviate the physical discomfort associated with these procedures. Several factors limit the strength of this study. These include the absence of data on participants' comorbidities, professional experience, monthly income levels, and the relatively small sample size. Additionally, average operating time and surgical experience for each prostatectomy technique were not recorded, which may have influenced the outcomes.

## CONCLUSION

This study found that the RALP technique was associated with lower levels of physical exhaustion compared to ORP and LRP. Although the impact of regular exercise on physical exhaustion and musculoskeletal complaints was modest, our findings suggest a potential benefit. We recommend promoting regular physical activity among urologists—especially those performing LRP and RALP—to help reduce physical strain.

Further large-scale studies are warranted to confirm these results and explore the broader implications of surgeon ergonomics and physical well-being in minimally invasive urologic surgery.

**Abbreviations**

| | |
|---|---|
| **BMI** | body mass index |
| **LRP** | laparoscopic radical prostatectomy |
| **ORP** | open radical prostatectomy |
| **RALP** | robot-assisted laparoscopic prostatectomy |
| **Q** | question |

## ACKNOWLEDGEMENTS

As authors, we would like to thank the Turkish Association of Urology, Turkish Endourology Society, and Turkurolap Education and Research Group for their support in reaching out to members of urologists.

### Funding
The authors received no funding for this work.

### Competing Interests
The authors declare there are no competing interests.

### Author Contributions
- Mahmut T. Olcucu conceived and designed the experiments, performed the experiments, analyzed the data, prepared figures and/or tables, and approved the final draft.
- Mustafa S. Bolat analyzed the data, prepared figures and/or tables, and approved the final draft.
- Kadir Yildirim performed the experiments, analyzed the data, authored or reviewed drafts of the article, and approved the final draft.
- Yasar Ozgok conceived and designed the experiments, performed the experiments, authored or reviewed drafts of the article, and approved the final draft.
- Theodoros Tokas analyzed the data, prepared figures and/or tables, authored or reviewed drafts of the article, and approved the final draft.
- Ali Gözen conceived and designed the experiments, performed the experiments, authored or reviewed drafts of the article, and approved the final draft.

### Ethics
The following information was supplied relating to ethical approvals (i.e., approving body and any reference numbers):

The Antalya Training and Research Hospital Ethics Kommittee approved the study (2019-023).

### Data Availability
Raw data is available in the Supplemental Files.

### Supplemental Information
Supplemental information for this article can be found online at http://dx.doi.org/10.7717/peerj.19908#supplemental-information.

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
