# Peer review of "Physical and mental effects of different radical prostatectomy techniques on urologic surgeons"

_PeerJ, doi:10.7717/peerj.19908_

## Round 0.1 · original submission · Major Revisions

Please attend to all the comments from each of the reviewers.

Reviewer 1 ·

Basic reporting

I congratulate the authors on preparing this manuscript, which mainly assesses the impact of different surgical techniques used for the radical prostatectomy procedure on surgeons' quality of life. The authors collected data from 160 urologists using online surveys distributed via various urological societies. Overall, it is a very interesting study and addresses an important topic—surgical ergonomics—highlighting points that could be addressed to prevent detrimental effects of surgeries and specific techniques on surgeons' quality of life. This guidance can help surgeons adopt preventive measures.
To further improve the manuscript, I have some suggestions for the authors:
1. The manuscript would significantly benefit from English language proofreading. There are quite a few grammatical errors in descriptions, such as in Lines 65, 66, 104, 107, 108, 122, 123, 124, etc., and spelling errors such as in Lines 68 and 202.
2. The effect of exercise on surgeons' musculoskeletal strength and operating room ergonomics has been studied previously, and such studies could be mentioned in the introduction. For instance, please see: Winters JN, et al. "Stretching and Strength Training to Improve Postural Ergonomics and Endurance in the Operating Room." Plast Reconstr Surg Glob Open. 2020;8(5):e2810. doi: 10.1097/GOX.0000000000002810. PMID: 33133890; PMCID: PMC7572150.
3. Authors should clarify why they add patients, patients’ families, and employers as potential stressors, especially in the context of ergonomics, and why this is hypothesized to differ between techniques.

Experimental design

Several improvements could be made to the Methods section:
a. By adding the preset criteria for including and excluding survey responses. Did all participants complete all survey questions without missing any? If there were survey responses missing questions, what were the preset criteria to exclude that particular response? For example, if a participant missed >=5 questions?
b. Survey questions described in the Methods section are unclear. Authors should either briefly describe the survey questions in a clearer way and cite the tables/supplementary files.
c. Authors should mention that the VAS scale was used.

Validity of the findings

In the Results section:
a. Authors report that 160 urologists responded to their survey, however, the numbers in Lines 119 and 120 do not sum to 160; authors should rephrase the sentence to express that these numbers represent the amount of cases completed annually.
b. Authors report that 102 participants used only one technique. What about the remaining 58 participants? Did they use multiple techniques? If yes, how did authors decide which patients should be included in which arm? Was this based on the preferred procedure types?
c. Lines 125-126 should clarify that this sentence refers to patient characteristics.

Additional comments

Discussion:
a. Again, as mentioned above in the Introduction section comments, authors mention work-employer, patient, and caregiver-related stress put on surgeons as a significant stressor. Authors must describe how they suggest preventing/dealing with the consequences of these conditions. Additionally, describing this as 'caregivers' not 'patient relatives' would be more appropriate in a scientific context.
b. Lines 136-137: Major surgeries using minimally invasive techniques… -> This statement does not convey the message correctly. In the modern era, new surgeons are trained primarily with minimally invasive techniques and new technologies for major surgeries from the early stages of their training. If authors mean the beginning of the learning curve for surgeons particularly experienced in open techniques compared to minimally invasive techniques, that should be clarified.
c. Lines 143-144 should be elaborated further or refined, as it is known that conventional laparoscopic procedures can cause surgeon discomfort and are not ergonomically optimal. However, for certain procedures, including radical prostatectomy, it is well known that the robotic approach provides better vision, access, and maneuvering in the pelvic cavity in experienced hands, and the ergonomics of robots such as the da Vinci Xi robotic console are much better.
d. Lines 186-193: The same sentences are repeated multiple times and this should be fixed.
e. Lines 197-199: The sentence regarding slight correlation—authors should provide the statistics or cite the table/supplementary material/figure here so that it is clear to the readers.
f. It should be clearly stated in the limitations section that average surgical/OR time per technique for each participant and surgeons’ experience was not captured; ideally, the results should have been adjusted for these factors.
g. Table 1 is shown twice; please fix this.
h. Regarding the eye strain, it appears there is no difference in medians. Could the authors elaborate on how many participants responded to this question?
i. Authors should also describe which variables were analyzed using parametric vs. non-parametric tests due to distribution/normality briefly in the table, for example, asterisks could be used to demonstrate this.

·

Basic reporting

The research study on the “Physical and mental effects of different radical prostatectomy techniques” by Mahmut Taha Olcucu et al. in 2024 should be improved to avoid excess repetition, clarity, flow, organization, grammar, and syntax. Some examples where the language and structure could be improved include lines 61-63, 65, 66, 68, 74, 81, 85, 96, 97, 118, 120, 121, 138, 141, 145, 148, 149, 150, 153, 164, 182, 187, 196, 197, 202. Line 187 is repeated 3 times (lines 188-193). Suggestions for the results section include numbering consistency. I suggest you have a colleague who is proficient in English and familiar with the subject matter review your manuscript, or contact a professional editing service. Organize by importance and be concise; do not repeat statements.

Experimental design

-While the primary objective of the article was to evaluate physical exhaustion as well as patient, relatives, and employer-related mental stress in surgeon performance in RP, it appears that the study's second objective was to know if physical activity exerts any benefits on surgeons' musculoskeletal complaints performing radical prostatectomies. I suggest adding this as a second objective.
-The hypothesis are partially correlated with the objectives of the study; researchers hypothesized that robotic-assisted laparoscopic prostatectomy (RALP) entails less physical exhaustion and musculoskeletal complaints over the other approaches while dismissing the mental stress caused to surgeons by patients, relatives, and patients' employers. The second hypothesis included that regular physical activity would decrease musculoskeletal complaints, though this was not part of the study objective.
-Researchers did not specify how the email addresses were obtained aside from social media.
-Question number 2 (“¿Before the radical prostatectomy, do your patients and/or their family (relatives) cause stress on you?”) could have been split into two separate questions to better evaluate stress caused by patients and relatives.

Validity of the findings

The statistical analysis is well-structured and appropriate for handling both parametric and non-parametric data. Statistical measures are appropriately included, giving credibility to the findings.

Additional comments

The introduction could emphasize the knowledge gap regarding the study's objective. The discussion section could be more concise.

Reviewer 3 ·

Basic reporting

I appreciate the opportunity to review the manuscript. The topic addressed in this study—evaluating the musculoskeletal impact on surgeons performing different prostatectomy techniques—is highly relevant. Additionally, the association between surgical outcomes and physical activity is an interesting and important angle. However, the manuscript presents some significant issues, as outlined below:

Typographical Errors:
- Line 68: "compared."
- Line 138: "controlling."
- Line 202: "showed."

Repetitive Content:
- Lines 185–193: The same phrase is repeated three times and requires revision for clarity and conciseness.

Experimental design

The use of online surveys for data collection offers a practical and efficient method for gathering information.
The study's focus on open prostatectomy as the predominant approach does not align with the practices in leading prostate cancer treatment centers, where robotic-assisted surgery is the standard. Future research should aim to include participants from these high-volume centers to ensure broader applicability and relevance.

Validity of the findings

The small sample size likely contributed to the lack of significant findings related to physical activity.
Although some studies suggest greater discomfort during robotic surgery, it is crucial to emphasize the importance of proper surgeon console positioning to optimize ergonomic outcomes.

---

## Round 0.2 · Major Revisions

**Language Note:** The review process has identified that the English language must be improved. PeerJ can provide language editing services - please contact us at [email protected] for pricing (be sure to provide your manuscript number and title). Alternatively, you should make your own arrangements to improve the language quality and provide details in your response letter. – PeerJ Staff

Reviewer 1 ·

Basic reporting

Please see the attached document.

Experimental design

Please see the attached document.

Validity of the findings

Please see the attached document.

Additional comments

Please see the attached document.

Annotated reviews are not available for download in order to protect the identity of reviewers who chose to remain anonymous.

·

Basic reporting

Overall, the authors did a good job revising and improving the manuscript. However, a few grammatical errors still need to be corrected, such as:
-Many "the" articles are missing.
-A missing space in line 106.
-The phrase in lines 105 and 106 is repeated 3 times (206,207, 209,210).
-A missing period in Line 217.
The article is now better structured and will provide valuable insights regarding how ergonomics and external factors are affecting the existing surgical techniques.

Experimental design

No comment

Validity of the findings

No comment

Reviewer 3 ·

Basic reporting

I would like to commend the authors for the thoughtful revisions made to the manuscript, which have significantly improved the clarity and quality of the text. Moreover, the focus on evaluating the physical and mental impact of different radical prostatectomy techniques, particularly regarding surgeon ergonomics, is highly commendable. Addressing the well-being of professionals involved in surgical procedures is a timely and important initiative, especially given the limited attention this topic often receives in medical literature. Studies like this are essential for raising awareness and promoting strategies to enhance the working conditions of surgeons, ultimately benefiting both healthcare professionals and patient outcomes.

Experimental design

No comment

Validity of the findings

No comment

Additional comments

This line of research is not only timely but also highly impactful, as it promotes a more sustainable and human-centered approach to surgical practice. There is great potential to expand this work and contribute even further to the understanding of occupational health in surgery.

---

## Round 0.3 · accepted · Accept

Please confirm the comments from a reviewer.

·

Basic reporting

The article has been revised and is now improved. Two small revisions; there are two extra spaces in the abstract that should be corrected, in the word "only" and in the word "preferred."

Experimental design

-

Validity of the findings

-